# Lifestyle Changes and Baseball Activity among Youth Baseball Players before and during the First COVID-19 Pandemic in Japan

**DOI:** 10.3390/children9030368

**Published:** 2022-03-07

**Authors:** Ryota Kuratsubo, Masashi Kawabata, Emi Nakamura, Masumi Yoshimoto, Satoshi Tsunoda, Yuji Takazawa, Hiroyuki Watanabe

**Affiliations:** 1Department of Rehabilitation, Kitasato Institute Hospital, Kitasato University, Tokyo 108-8642, Japan; r.kura@kitasato-u.ac.jp or; 2Department of Sports Medicine, Juntendo University Graduate School of Medicine, Tokyo 113-8421, Japan; ytakaza@juntendo.ac.jp; 3Department of Rehabilitation, School of Allied Health Sciences, Kitasato University, Sagamihara 252-0373, Japan; hw@ahs.kitasato-u.ac.jp; 4Department of Physical Therapy, Faculty of Health Science, Juntendo University, Tokyo 113-8421, Japan; e.nakamura.wu@juntendo.ac.jp; 5Department of Physical Therapy, Faculty of Health and Medical Science, Teikyo Heisei University, Tokyo 170-8445, Japan; m.yoshimoto@thu.ac.jp; 6Department of Rehabilitation, Shonan Fujisawa Tokushukai Hospital, Fujisawa 251-0041, Japan; tsunoda0707@gmail.com

**Keywords:** COVID-19, lifestyle changes, adolescents, baseball, exercise, screen time

## Abstract

This cross-sectional study aimed to clarify the changes in lifestyle and baseball activity before and during the COVID-19 pandemic among youth baseball players. Participants were 99 youth baseball players (ages 9.6 ± 1.5 years, height 137.8 ± 9.4 cm, weight 35.3 ± 12.4 kg) in Japan. They completed an online survey between April 6 and 20, 2021, on their demographic characteristics, lifestyle (recreational screen, sleep, and study times), and baseball activity (frequency of team practice time and voluntary exercise-related baseball) at two-time points: before the pandemic (before March 2020) and during the state of emergency declared in Japan (from April to May 2020). The changes in outcomes between the two-time points were evaluated for significance. Recreational screen time and sleeping time during the state of emergency were significantly increased compared to those before the pandemic (*p* < 0.001). The frequency of team practice time on weekends during the state of emergency was significantly reduced, and voluntary exercise-related baseball was significantly increased compared to that before the pandemic (*p* < 0.01). We found that the COVID-19 pandemic changed behaviors concerning activities and exercise among youth baseball players and recommended that such behavioral changes be carefully monitored.

## 1. Introduction

The coronavirus disease 2019 (COVID-19) pandemic, declared by the World Health Organization on 11 March 2020, disrupted daily activities due to the need for social isolation. In Japan, the first state of emergency was declared by the Japanese government from 7 April to 31 May 2020. This declaration resulted in restrictions on the activities of Japanese citizens, and it has been reported that the amount of time spent engaging in physical activity decreased by 26.5% during the state of emergency (April 2020) compared with a pre-COVID-19 period (January 2020) among older adults in Japan [1].

Youth lifestyle behavior has also changed because of the COVID-19 pandemic. School closures during the COVID-19 pandemic were one of the major changes that affected the youth. The school closures made youth study at their home instead of going to school, and they lost the regularity of getting up and going to the school at a certain time. It was also possible that the loss of time spent commuting to school is causing them to sleep longer. Additionally, children lost access to opportunities for physical activity and sports outside of school, and the time spent at home increased. As a result, it has been reported that physical activity decreased, and screen time increased among pre-adolescent and adolescents (6–17 years old) in Shanghai [2] and children with obesity living in Italy [3]. A decrease in physical activity and an increase in screen time were observed in young people aged 13–19 years in Australia [4]. Reduced physical activity and prolonged screen time are associated with negative physical and mental health outcomes, such as loss of muscular and cardiorespiratory fitness, obesity, high blood pressure, and insulin resistance [5], as well as depression, anxiety, suicide, and inattention [5,6]. It is also known that a lack of physical activity and increased screen time result in poor academic achievements [7] and poor sleep [5,8]. Therefore, youth lifestyle behavior should be focused on screen time, study time, and sleeping time in the lifestyle.

Youth athletes, in particular, may have been negatively affected by the cancellation of school and sports [9]. A previous study reported that, among youth athletes, the sleeping time was higher, and the frequency of practice was lower during the pandemic compared with before the pandemic [10]. However, few studies have examined the negative effects of the COVID-19 pandemic on youth athletes in specific sports. Baseball, played by more than 8 million individuals, is the most popular sport in Japan, followed by soccer (6.7 million) and basketball (4.8 million) [11]. Youth sports offer several social, physical, and mental health benefits for adolescents [12,13]. In particular, the state of emergency has made it difficult for youth baseball players to practice and play games due to the closure of baseball fields and the voluntary restraint exercised concerning team activities. However, few studies have investigated the effects of the pandemic on activity and exercise levels among youth baseball players. This study aimed to clarify the changes in lifestyle and baseball behavior in youth baseball players in Japan before and during the COVID-19 pandemic.

## 2. Method

### 2.1. Design and Participants

This study was conducted in accordance with the Checklist for Reporting Results of Internet E-Surveys (CHERRIES) for reporting web-based surveys [14]. A cross-sectional survey of youth baseball players aged 6–12 years old was conducted using a web-based questionnaire (Google Form^®^ by Google LLC, Los Angeles, CA, USA). Two of the authors (E.N. and M.Y.) developed the original web-based questionnaire for this study, and usability and technical functionality were tested by four of the authors (R.K., M.K., E.N., and M.Y). Informed consent was obtained from all study participants through the website prior to answering the questionnaire. The 77 baseball teams registered with the Junior Baseball Federation of Sagamihara City, Kanagawa Prefecture, Japan, were initially approached to participate in this study. First, the staff of the Junior Baseball Federation of Sagamihara City sent an e-mail with the URL of the web-based questionnaire to the officials of the 77 teams. Next, when the team leaders agreed to participate, the e-mails were forwarded from the team officials to the players on each team. This web-based questionnaire was administered from April 6 2021 to April 20 2021, and the parents of the participants were asked to help their children complete the questionnaire. The time required for a participant to complete the survey was approximately 15 min. This was a closed survey comprising 26 items over 3 pages. Participation in the study was voluntary, and no rewards were given to the participants for completing the survey. Neither randomization of items nor adaptive questioning methods were used in this survey.

We excluded those who did not agree to participate in the study, those who were not registered with a team between January and March 2020, or those responses were incomplete or had missing values.

### 2.2. Survey Items

In this study, ‘before the pandemic’ was defined as before March 2020, and ‘during the state of emergency’ was defined as the period from April to May 2020. Survey items included demographic characteristics (age, sex, height, and weight), lifestyle, and baseball activity.

### 2.3. Lifestyle

The lifestyle questionnaire consisted of 3 questions on recreational screen time, study time, and sleeping time. Participants were asked about the approximate time spent as recreational screen time (RST) [15]. This included watching television, playing games on any device, and using the Internet for recreational use on weekdays and weekends. They were asked about their estimated study time at their home on weekdays and weekends. They were then asked about their estimated sleeping time, calculated from bedtime to wake-up time. Finally, the total sleep time was calculated.

### 2.4. Baseball Activity

We asked about the frequency of participation in team practice time and voluntary exercise-related baseball during both weekdays and weekends. The players were asked to select the 3 most frequent voluntary practices from catching a ball, throwing at a wall, swinging, batting center practice, and running for more than 15 min.

### 2.5. Statistical Analysis

The valid response rate was calculated from the ratio of the number of valid responses (A) to the number of players registered in teams that cooperated in this study (B) (A/B × 100, unit: %). The RST and study times were categorized into 4 time periods: <1 h, 1–2 h, 2–3 h, and ≥3 h. The American Academy of Pediatrics has suggested specific limits on screen time (< 1–2 h per day) for entertainment purposes [16]. The participants were divided into 2 groups: short-RST (<2 h) and long-RST (≥2 h). The frequency of team practice time was categorized as: none, <3 h, 3–5 h, and ≥5 h. The frequency of voluntary exercises was categorized as: none, 1–3 times/week, 4–5 times/week, and 6–7 times/week, and the assessment of voluntary exercise consisted of throwing (playing catch, throwing at the wall, other types of throwing), batting (swing practice, batting center practice, other batting items), running (≥15 min, other running items), and none.

Continuous variables were expressed as mean and standard deviations and were evaluated using the student’s t-test or the Mann–Whitney U test. Categorical data were expressed as number and percentage (%) and were evaluated using the chi-squared test. We used paired t-tests or the Wilcoxon test to evaluate significant changes between the period prior to the pandemic (March 2020) and during the state of emergency (April to May 2020) in the item responses. The statistical test level was set to *p* < 0.05. SPSS software (version 26.0, IBM Corp., Armonk, NY, USA) was used to perform all the tests.

## 3. Results

The number of teams that participated in this study was 33 out of the maximum possible 77 (42.9%). A questionnaire was sent to 538 players belonging to these 33 teams, and responses were received from 103 players. Data from 99 players who did not meet the exclusion criteria were analyzed (valid response rate, 18.4%). In terms of the sleeping time, three players were excluded because of missing values, with 96 players included concerning this item. The players included seven female adolescents (7.1%). They had a mean age of 9.6 ± 1.5 years, a mean height of 137.8 ± 9.4 cm, and a mean weight of 35.3 ± 12.4 kg.

The lifestyle results are presented in Table 1. During the state of emergency, short-RST (<2 h) reduced, and long-RST (≥2 h) increased compared to the values before the pandemic. Study time at home on a weekday during the state of emergency significantly increased compared to study time before the pandemic. On the other hand, study time on weekends did not significantly change from before the pandemic compared with during the state of emergency. Sleeping time during the state of emergency was significantly longer than before the pandemic.

Regarding baseball activity, the frequency of weekend team practice time during the state of emergency was significantly shorter than that before the pandemic, although the frequency of team practice time on weekdays did not significantly change from that before the pandemic (Table 2 and Figure 1). The frequency of voluntary exercise during the state of emergency was higher than that before the pandemic (Table 3, *p* = 0.019). The rate of voluntary exercise undertaken 6–7 times weekly before the pandemic was only 9.1%, but during the state of emergency, it increased to 20.2%.

The type of items involved in voluntary exercise significantly changed (*p* = 0.01), with playing catch and running for ≥15 min increased whereas playing at the batting center was reduced (Table 4).

## 4. Discussion

We investigated the impact of the COVID-19 pandemic on activity and exercise levels among youth baseball players. Our study is the first to clarify changes in behavior among youth baseball players due to social regulations imposed during the pandemic. We found that RST was significantly increased and that the frequency of team practice decreased during the state of emergency compared with the situation before the pandemic.

### 4.1. Lifestyle

RST during the state of emergency was significantly longer than that before the pandemic (Table 1). In particular, the rate of long-RST (≥2 h) during the state of emergency increased by approximately 200–250% compared to that before the pandemic. The American Academy of Pediatrics has recommended < 2 h of RST for youth [15]. It has been reported that screen time increased by 4.85 h per day in pre-adolescent and adolescent children with obesity living in Italy [3]. As in our study, a high rate of long-RST (69.5 % on weekdays and 63.8 % on weekends) was observed in European children during the COVID-19 pandemic [17].

On the other hand, the rate of study hours at home of <1 h was not significantly different between the period before the pandemic and during the state of emergency (Table 1). In addition, the sleeping time during the state of emergency was significantly longer than that before the pandemic, although the change in sleeping time was only 8 min. Youth athletes in the United States reported significantly increased average sleeping time per night, from 7.9 h before the pandemic to 9.4 h during the pandemic [10]. Notably, the National Sleep Foundation in America has recommended 9–11 h of sleep daily for school-aged children [18]. In a study with 160 student athletes, those with less than eight hours of sleep per night were reportedly 1.7 times more likely to sustain a musculoskeletal injury [19]. The reason for the difference in the changes in sleeping time between the previous study and this study may be that the participants of this study routinely slept for 9 h per night even before the pandemic, which was longer than youth athletes in the United States. Moreover, there would be no reason to prolong the sleeping time because it was already optimal even before the pandemic. Further prospective cohort studies are necessary to investigate the lifestyle changes, especially prolonged RST, which affects the physical and mental conditions among youth baseball players.

### 4.2. Baseball Activity

The amount of team practice time on weekends during the state of emergency significantly decreased compared to that before the pandemic, although the practice time on weekdays was not significantly different between the period before the pandemic and during the state of emergency (Table 2, Figure 1). Baseball practice and game time before the pandemic usually took place on weekends for a period ≥ 5 h in Japan; thus, the COVID-19 pandemic affected baseball activity on weekends among youth baseball players. Two-thirds of the baseball teams canceled team practices, with one-third of the teams continuing to practice. We did not know the reason why these baseball teams canceled team practice because we did not collect data regarding the underlying reason. However, the answers to the questionnaire of this study were collected during the first state of emergency. This was the period when the Japanese people were confused by the high level of infection and its high mortality rate, and the Japanese government ordered school closures and restricted access to sports centers, playgrounds, and public parks. Therefore, it was not surprising that team practice was canceled. Even for the teams that had been practicing, the practices were arranged as short programs for a duration of 3 h with infection control. Youth athletes in the United States reportedly trained for 9.7 h per week on average before the pandemic. This number reduced significantly to 6.4 h per week during the pandemic [10]. Although we could not directly compare the training time because such information was unavailable in our study, we posit that the lack of team practice may have detrimentally influenced the youth baseball players. Because baseball is a team sport, throwing and practicing in the bullpen requires at least one other player. In actuality, the incidence of injuries among Major League Baseball players after the COVID-19 pandemic was almost double of that before the pandemic; the increase in the injury incidence was seen in the upper extremities, spine/core, and other injuries, and not the lower extremities [20]. One of the reasons for the increased injury was the insufficient throwing practice time [20] and throwing in fewer preseason innings [21]. On the other hand, the excessive number of throws [22] and training hours per week [23] were also reported as one of the factors of baseball-related pain in the shoulder and elbow among youth baseball players. Therefore, further investigation is required to clarify whether these changes in baseball team practice time after the pandemic was a factor in the increased incidence of injuries, especially in the upper extremities, among youth baseball players.

On the other hand, the frequency of voluntary exercise undertaken 6–7 times weekly ranged from 9.1% before the pandemic to 20.2% during the state of emergency (Table 3). In particular, the running rate doubled (Table 4). Playing catch and batting practice were the items that are most frequently engaged in, with no change occurring between the period before the pandemic and during the state of emergency. The amount of throwing at the wall decreased, whereas the number of those playing catch increased. In addition, the number of participants who went to the batting center decreased by half. These findings may have been because many teams could no longer practice, a small number of volunteers gathered to practice baseball, and the batting center was often closed due to COVID-19 infection control. Among Major League Baseball players, there were no significant differences in the incidence of injury to the lower extremity between the period before and during the pandemic, which was thought to be the result of isolated exercises and weight training using body-weight muscle strength without any fitness equipment/facilities [20]. In our study, we could not clarify the effect of voluntary exercise; however, we expect voluntary exercise is largely beneficial in terms of promoting physical activities, skill training, and prevention of injuries, especially in the lower extremities, among youth baseball players.

### 4.3. Limitations

Our study has several limitations. First, there can be selection bias in web surveys [24] and exercise frequency data may be subject to recall bias, which may have led to a reduced valid response rate from youth baseball players. Moreover, the data were collected from only one city in Japan. Second, this was a small cross-sectional study that reported only preliminary results. Third, further longitudinal studies are needed to observe changes as participants grow older, and related developments occur in the techniques of baseball play. Finally, data regarding the lifestyle other than those discussed in this study and participants’ physical activities that were not related to baseball were not collected, and differences in other lifestyle factors and non-baseball activity during the state of emergency compared to before the pandemic were not analyzed in this study.

## 5. Conclusions

This study investigated lifestyle and baseball activity changes among youth baseball players due to the impact of the COVID-19 pandemic during the state of emergency declared in Japan between April and May 2020. We found that RST was significantly increased during the state of emergency and that baseball team practice decreased compared to the period before the pandemic. From these results, it is clear that the COVID-19 pandemic led to behavioral changes concerning activities and exercise among youth baseball players. Longitudinal studies are needed that consider changes occurring as players mature and related developments in the techniques of baseball play. The results and conclusions of this investigation may be considered a basis and starting point for further research.

## Figures and Tables

**Figure 1 children-09-00368-f001:**
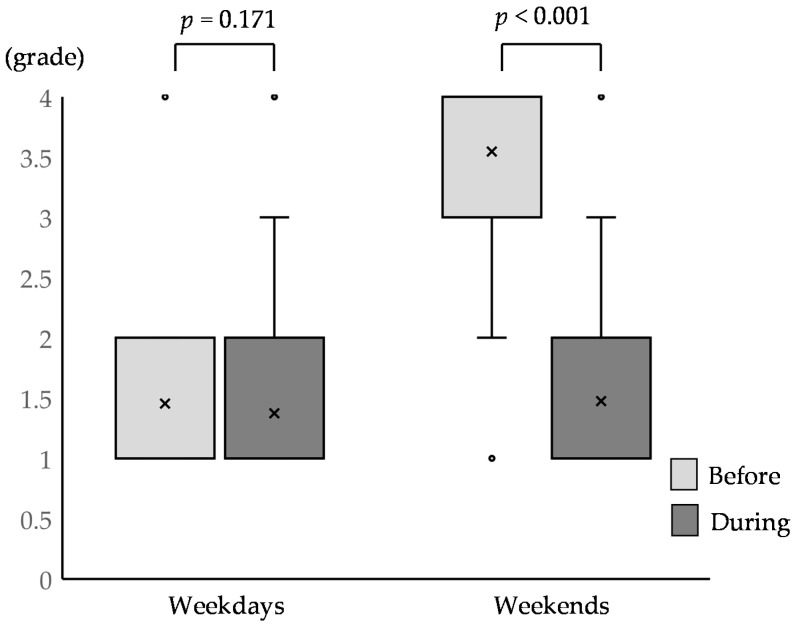
Frequency changes in team practice on weekdays and weekends (*n* = 99). Data are shown as median values and calculated using the Wilcoxon test. ‘×’ demonstrate the mean value and ‘∘’ demonstrate the outlier. Abbreviations: before, before the COVID-19 pandemic; during, during the first declaration of the state of emergency; grade 1, none; grade 2, <3 h; grade 3, 3–5 h; grade 4, >5 h.

**Table 1 children-09-00368-t001:** RST and study hour changes at home on weekdays and weekends (*n* = 99).

		Weekdays					Weekends					
		Before		During		*p*-Value	Before		During		*p*-Value	
RST, % (number)	<1 h	45.5%	(45)	35.4%	(35)		50.5%	(50)	27.3%	(27)		
	1–2 h	35.4%	(35)	17.2%	(17)		27.3%	(27)	19.2%	(19)		
	2–3 h	10.1%	(10)	12.2%	(18)		10.1%	(10)	14.4%	(14)		
	≥3 h	9.1%	(9)	29.3%	(29)		12.1%	(12)	39.4%	(39)		
	Grade median (IR; 25% to 75%)	2	(1 to 2)	2	(1 to 4)	<0.001	1	(1 to 2)	3	(1 to 4)	<0.001	
	Short (<2 h/day)	80.9%	(80)	52.6%	(52)	<0.001	77.8%	(77)	46.5%	(46)	<0.001	
	Long (≥2 h/day)	19.2%	(19)	41.5%	(47)		22.2%	(22)	53.8%	(53)		
Study hours, % (number)	<1 h	92.9%	(92)	77.8%	(77)		90.9%	(90)	85.9%	(85)		
	1–2 h	5.1%	(5)	16.7%	(16)		8.1%	(8)	12.1%	(12)		
	2–3 h	0.0%	(0)	3.0%	(3)		0.0%	(0)	0.0%	(0)		
	≥3 h	2.0%	(2)	3.0%	(3)		1.0%	(1)	2.0%	(2)		
	Grade median (IR; 25% to 75%)	1	(1 to 1)	1	(1 to 1)	<0.001	1	(1 to 1)	1	(1 to 1)	0.109	
		**Before**					**During**				***p*-value**	
Sleeping time (*n* = 96), min (SD)		560.0 (37.2)					568.8 (42.7)				0.006	

Grade 1: <1 h/day, Grade 2: 1–2 h/day, Grade 3: 2–3 h/day, Grade 4: ≥3 h/day. Abbreviations: RST, recreational screen time; before, before the COVID-19 pandemic; during, during the declaration of the state of emergency; h, hour(s); IR, interquartile range.

**Table 2 children-09-00368-t002:** Frequency changes in team practice time on weekdays and weekends (*n* = 99).

		Weekdays					Weekends				
		Before		During		*p*-Value	Before		During		*p*-Value
The frequency of the practice time,% (number)	None	62.6%	(62)	65.7%	(65)		5.1%	(5)	60.6%	(60)	
	<3 h	33.3%	(33)	32.3%	(32)		5.1%	(5)	33.3%	(33)	
	3–5 h	0.0%	(0)	1.0%	(1)		20.2%	(20)	4.0%	(4)	
	≥5 h	4.0%	(4)	1.0%	(1)		69.7%	(69)	2.0%	(2)	
	Grade median (IR; 25% to 75%)	1	(1 to 2)	1	(1 to 2)	0.171	4	(3 to 4)	1	(1 to 2)	<0.001

Grade 1: None, Grade 2: <3 h/day, Grade 3: 3–5 h/day, Grade 4: ≥5 h/day. Abbreviations: before, before the pandemic; during, during the 1st pandemic; h, hour(s); IR, interquartile range.

**Table 3 children-09-00368-t003:** Frequency changes in voluntary exercise-related baseball on weekdays and weekends (*n* = 99).

		Before		During		*p*-Value
The frequency of the voluntary exercise, % (number)	None	24.2%	(24)	26.3%	(26)	
	1–3 times/week	46.5%	(46)	32.3%	(32)	
	4–5 times/week	20.2%	(20)	21.2%	(21)	
	6–7 times/week	9.1%	(9)	20.2%	(20)	
	Grade median (IR; 25% to 75%)	2	(2 to 3)	2	(1 to 3)	0.019

Grade 1: None, Grade 2: 1–3 times/week, Grade 3: 4–5 times/week, Grade 4: 6–7 times/week. Abbreviations: before, before the pandemic; during, during the 1st pandemic; h, hour(s); IR, interquartile range.

**Table 4 children-09-00368-t004:** Item changes in voluntary exercise-related baseball (*n* = 99).

		Before		During		*p*-Value
Throwing %; number	Playing catch	17.2%	(51)	21.2%	(63)	0.01
	Throwing on the wall	6.1%	(18)	2.4%	(7)	
	Other throwing items	1.0%	(3)	1.3%	(4)	
Batting %; number	Swing practice	19.9%	(59)	21.9%	(65)	
	Batting center	8.8%	(26)	4.4%	(13)	
	Other batting items	1.3%	(4)	3.4%	(10)	
Running %; number	Running more than 15 min	3.4%	(10)	7.7%	(23)	
	Other running items	0.3%	(1)	1.0%	(3)	

## Data Availability

The data presented in this study are available on request from the corresponding author.

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
