# Peer review of "Lifestyle Changes and Baseball Activity among Youth Baseball Players before and during the First COVID-19 Pandemic in Japan"

_children, 2022, doi:10.3390/children9030368_

Round 1

Reviewer 1 Report

Overall speaking, this paper is quite descriptive in nature and has demonstrated the detraining patterns of the youth baseball players and their lifestyle changed during the pandemic.

The statistical analysis is presented systematically.

However, in the discussion session, it would be better to have some recommendations to tackle the health problem identified in this study, such as how to promote physical activity/skill training or other training mode for the youth baseball players.

Author Response

Response to Reviewer 1 Comments

Point 1 : Overall speaking, this paper is quite descriptive in nature and has demonstrated the detraining patterns of the youth baseball players and their lifestyle changed during the pandemic. The statistical analysis is presented systematically.

Response 1: Thank you very much for your comment.

Point 2 : However, in the discussion session, it would be better to have some recommendations to tackle the health problem identified in this study, such as how to promote physical activity/skill training or other training mode for the youth baseball players.

Response 2: In accordance with your comment, we have added the following text to the Discussion (Page 7-8, Lines 251–258):

” Among Major League Baseball players, there were no significant differences in the incidence of injury to the lower extremity between the period before and during the pandemic, which was thought to be the result of isolated exercises and weight training using body-weight muscle strength without any fitness equipment/facilities [20]. In our study, we could not clarify the effect of voluntary exercise; however, we expect voluntary exercise is largely beneficial in terms of promoting physical activities, skill training, and prevention of injuries, especially in the lower extremities, among youth baseball players.”

We have also added Reference number 20:

Platt, B.N.; Uhl, T.L.; Sciascia, A.D.; Zacharias, A.J.; Lemaster, N.G.; Stone, A.V. Injury Rates in Major League Baseball During the 2020 COVID-19 Season. Orthop. J. Sports Med. 2021, 9, 2325967121999646. DOI:10.1177/2325967121999646.

Reviewer 2 Report

Thank you for possibility of reviewing article. Idea of research is good according to popularity of this sport discipline in Japan. Composition of article is clear, I suggest to extend number of references with last released in similar (COVID-19) subject.

Questionable is number of respondents – 99 due to 77 clubs, what give considerations about reliability of research. Line 119 to 122, assume that response of questionaries was at  18,6%, but this number is from 33/77 clubs, so in meaning of population of youth playing baseball is much lower. 

Table 1. There is mistake in weekends - two times during instead of before and during. 

It would be more interesting for continuation of your research to extend questionaries in more specific questions about RST etc., due to obvious spending more time according to e-learning. 

Author Response

Response to Reviewer 2 Comments

Point 1:Thank you for possibility of reviewing article. Idea of research is good according to popularity of this sport discipline in Japan. Composition of article is clear, I suggest to extend number of references with last released in similar (COVID-19) subject.

Response 1: Thank you very much for your comment. In accordance with your comment, we have added the following text to the Introduction (Page 2, Lines 63–66):

” A previous study reported that, among youth athletes, the sleeping time was higher, and the frequency of practice was lower during the pandemic compared with before the pandemic [10]. However, few studies have examined the negative effects of the COVID-19 pandemic on youth athletes in specific sports.”

Additionally, we have added the following paragraphs to the Discussion:

(Page 7, Lines 225–235)

“Youth athletes in the United States reportedly trained for 9.7 hours per week on average before the pandemic; this number reduced significantly to 6.4 hours per week during the pandemic [10]. Although we could not directly compare the training time because such information was unavailable in our study, we posit that the lack of team practice may have detrimentally influenced the youth baseball players. Because baseball is a team sport, throwing and practicing in the bullpen requires at least one other player. In actuality, the incidence of injuries among Major League Baseball players after the COVID-19 pandemic was almost double of that before the pandemic; the increase in the injury incidence was seen in the upper extremities, spine/core, and other injuries, and not the lower extremities [20]. One of the reasons for the increased injury incidence was the insufficient throwing practice time [20] and throwing in fewer preseason innings [21].

We have also added Reference numbers 10, 20, and 21:

Ellis, H.B.; Ulman, S.M.; John Wagner, K.J.; Carpenter, C.M.; Gale, E.B.; Shea, K.G.; Wilson, P.L. Youth Athletes Sleep More, Practice Less, and May Lose Interest in Playing Sports Due to Social Distancing Mandates. Prev. Med. Rep. 2022, 26, 101722. DOI:10.1016/j.pmedr.2022.101722.

Platt, B.N.; Uhl, T.L.; Sciascia, A.D.; Zacharias, A.J.; Lemaster, N.G.; Stone, A.V. Injury Rates in Major League Baseball During the 2020 COVID-19 Season. Orthop. J. Sports Med. 2021, 9, 2325967121999646. DOI:10.1177/2325967121999646.

Paul, R.W.; Omari, A.; Fliegel, B.; Bishop, M.E.; Erickson, B.J.; Alberta, F.G. Effect of COVID-19 on Ulnar Collateral Ligament Reconstruction in Major League Baseball Pitchers. Orthop. J. Sports Med. 2021, 9, 23259671211041359. DOI:10.1177/23259671211041359.

Point 2: Questionable is number of respondents – 99 due to 77 clubs, what give considerations about reliability of research. Line 119 to 122, assume that response of questionaries was at  18,6%, but this number is from 33/77 clubs, so in meaning of population of youth playing baseball is much lower.

Response 2: The e-mail with the URL of the web-based questionnaire was sent only to 33 teams which were team leaders who agreed to participate in this study. If the e-mail had been sent to the other 44 teams, the number and rate of response might have been different from this result. In terms of the response rate in this study, we believe that the valid response rate of 18.6% is correct because the e-mail was sent to 538 people in 33 teams and there were 99 valid responses. Therefore, we do not have to revise the results regarding the response rate.

However, we have revised the text in the Design and Participants subsection of Method to make it easier for the reader to understand the method of sending the e-mail with the URL of the web-based questionnaire

From:

“The 77 baseball teams registered with the Junior Baseball Federation of Sagamihara City, Kanagawa Prefecture, Japan, were initially approached to participate in this study. We excluded those who did not agree to participate in the study or individuals who were not registered with a team between January and March 2020.”

To: (Pages 2-3, Lines 85–100)

“The 77 baseball teams registered with the Junior Baseball Federation of Sagamihara City, Kanagawa Prefecture, Japan, were initially approached to participate in this study. First, the staff of the Junior Baseball Federation of Sagamihara City sent an e-mail with the URL of the web-based questionnaire to the officials of the 77 teams. Next, when the team leaders agreed to participate, the e-mails were forwarded from the team officials to the players on each team. This web-based questionnaire was administered from April 6, 2021 to April 20, 2021, and the parents of the participants were asked to help their children complete the questionnaire. The time required for a participant to complete the survey was approximately 15 minutes. This was a closed survey, comprising of 26 items over 3 pages. Participation in the study was voluntary, and no rewards were given to the participants for completing the survey. Neither randomization of items nor adaptive questioning methods were used in this survey.

We excluded those who did not agree to participate in the study, those who were not registered with a team between January and March 2020, or those whose responses were incomplete or had missing values.”

We have also added Reference number 14:

Eysenbach, G. Improving the Quality of Web Surveys: The Checklist for Reporting Results of Internet E-Surveys (CHERRIES). J. Med. Internet Res. 2004, 6, e34. DOI:10.2196/jmir.6.3.e34.

Point 3: Table 1. There is mistake in weekends - two times during instead of before and during.

Response 3: In accordance with your comment, we have revised the value of sleeping time in Table 1.

Point 4: It would be more interesting for continuation of your research to extend questionaries in more specific questions about RST etc., due to obvious spending more time according to e-learning.

Response 4: Thank you very much for your comment. We agree with your comment, and I would like to add a question about e-learning in succeeding studies.

Reviewer 3 Report

The study presented may be of interest to the readers of the journal. In addition, it could be useful for public institutions to develop public health policies on the promotion of sports practice and the reduction of sedentary habits during the COVID-19 pandemic and after overcoming it. In any case, this is a small sample, so the results and conclusions of this investigation should be taken only as a starting point for further studies.

In relation to keywords, it is recommended to include "baseball" among them.

The introduction is clear, concrete, concise and it is well structured. However, it is necessary to base on more bibliographic references on similar studies conducted in other sports with similar characteristics to baseball.

The aim of the research is clear and relevant. In the concept of lifestyle, the analysis of physical activity during the state of emergency compared to before the pandemic is missing. It could be that the changes in the practice of baseball had to do with other physical and sports activities more appropriate to the situation generated by the pandemic.

The research design seems to fit the purpose of the research, the statistical analyses used are appropriate for the type of sample and the variables analysed. The Wilcoxon test is mentioned only in the note to Figure 1 and should also be included in Section 2.5 (Statistical Analysis).

The instruments used for data collection are described but the procedure followed for the online survey is not explained. Has a specific reference been taken as a reference or has it been developed ad hoc? For online questionnaires, it is recommended to include a reference. By way of example: Eysenbach, G. (2004). Improving the quality of web surveys: The Checklist for Reporting Results of Internet E-Surveys (CHERRIES). Journal of Medical Internet Research, 6, 1-6. http://dx.doi.org/10.2196/jmir.6.3.e34.

The results section is well structured and explained.

Bibliography should be added in the discussion. It is not sufficiently contrasted with other studies to give solidity to the conclusions.

The conclusions are presented clearly and are justified by the results presented. However, it is necessary to improve the discussion by including more references.

The authors must incorporate the doi in the line reference 263:

doi: 10.18053/jctres.06.202002.003

Author Response

Response to Reviewer 3 Comments

Point 1: The study presented may be of interest to the readers of the journal. In addition, it could be useful for public institutions to develop public health policies on the promotion of sports practice and the reduction of sedentary habits during the COVID-19 pandemic and after overcoming it.

Response 1: Thank you very much for your comment.

Point 2: In any case, this is a small sample, so the results and conclusions of this investigation should be taken only as a starting point for further studies.

Response 2: We agree with your comment and also considered the sample of this study to be small. Hence, we have described it as a limitation of this study in the manuscript. Furthermore, in accordance with your comment, we have added the following sentence to the Conclusion (Page 8, Line 279-281):

”The results and conclusions of this investigation may be considered a basis and starting point for further research.”

Point 3: In relation to keywords, it is recommended to include "baseball" among them.

Response 3: In accordance with your comment, we have added “baseball” to the Keywords. (Page 1, Line 35)

Point 4: The introduction is clear, concrete, concise and it is well structured. However, it is necessary to base on more bibliographic references on similar studies conducted in other sports with similar characteristics to baseball.

Response 4: In accordance with your comment, we have added the following text to the Introduction (Page 3, Lines 63–66):

”A previous study reported that, among youth athletes, the sleeping time was higher, and the frequency of practice was lower during the pandemic compared with before the pandemic [10]. However, few studies have examined the negative effects of the COVID-19 pandemic on youth athletes in specific sports.”

We have also added Reference number 10:

Ellis, H.B.; Ulman, S.M.; John Wagner, K.J.; Carpenter, C.M.; Gale, E.B.; Shea, K.G.; Wilson, P.L. Youth Athletes Sleep More, Practice Less, and May Lose Interest in Playing Sports Due to Social Distancing Mandates. Prev. Med. Rep. 2022, 26, 101722. DOI:10.1016/j.pmedr.2022.101722.

Point 5: The aim of the research is clear and relevant. In the concept of lifestyle, the analysis of physical activity during the state of emergency compared to before the pandemic is missing. It could be that the changes in the practice of baseball had to do with other physical and sports activities more appropriate to the situation generated by the pandemic.

Response 5: In this study, we did not analyze other physical and sporting activities because we did not collect data regarding these activities. Therefore, in accordance with your comment, we have added the following sentence to the Limitations (Page 8, Lines 267–270):

”Finally, data regarding the lifestyle other than those discussed in this study and participants’ physical activities that were not related to baseball were not collected, and differences in other lifestyle factors and non-baseball activity during the state of emergency compared to before the pandemic were not analyzed in this study.”

Point 6: The research design seems to fit the purpose of the research, the statistical analyses used are appropriate for the type of sample and the variables analysed. The Wilcoxon test is mentioned only in the note to Figure 1 and should also be included in Section 2.5 (Statistical Analysis).

Response 6: In accordance with your comment, we have added “or the Wilcoxon test” to the Statistical Analysis section. (Page 3, Line 135)

Point 7: The instruments used for data collection are described but the procedure followed for the online survey is not explained. Has a specific reference been taken as a reference or has it been developed ad hoc? For online questionnaires, it is recommended to include a reference. By way of example: Eysenbach, G. (2004). Improving the quality of web surveys: The Checklist for Reporting Results of Internet E-Surveys (CHERRIES). Journal of Medical Internet Research, 6, 1-6. http://dx.doi.org/10.2196/jmir.6.3.e34.

Response 7: Thank you very much for your comment and the introduction to the CHERRIES checklist.

In accordance with your comment and following the guidelines of the checklist, we revised the text in the Design and Participants subsection of Method

From:

“A cross-sectional survey of youth players aged 6–12 years old was conducted using a web-based questionnaire (Google Form® by Google LLC, Los Angeles, CA, USA) from April 6, 2021, to April 20, 2021.Informed consent was obtained from all study participants prior to .

The 77 baseball teams registered with the Junior Baseball Federation of Sagamihara City, Kanagawa Prefecture, Japan, were initially approached to participate in this study. We excluded those who did not agree to participate in the study or individuals who were not registered with a team between January and March 2020. In addition, the parents of the children were asked to help their children to complete the questionnaire. Informed consent was obtained from all study participants prior to the study.”

To: (Pages 2-3, Lines 78–100)

“This study was conducted in accordance with the Checklist for Reporting Results of Internet E-Surveys (CHERRIES) for reporting web-based surveys [14]. A cross-sectional survey of youth baseball players aged 6–12 years old was conducted using a web-based questionnaire (Google Form® by Google LLC, Los Angeles, CA, USA). Two of the authors (E.N. and M.Y.) developed the original web-based questionnaire for this study, and usability and technical functionality were tested by four of the authors (R.K., M.K., E.N., and M.Y). Informed consent was obtained from all study participants through the website prior to answering the questionnaire. The 77 baseball teams registered with the Junior Baseball Federation of Sagamihara City, Kanagawa Prefecture, Japan, were initially approached to participate in this study. First, the staff of the Junior Baseball Federation of Sagamihara City sent an e-mail with the URL of the web-based questionnaire to the officials of the 77 teams. Next, when the team leaders agreed to participate, the e-mails were forwarded from the team officials to the players on each team. This web-based questionnaire was administered from April 6, 2021 to April 20, 2021, and the parents of the participants were asked to help their children complete the questionnaire. The time required for a participant to complete the survey was approximately 15 minutes. This was a closed survey, comprising of 26 items over 3 pages. Participation in the study was voluntary, and no rewards were given to the participants for completing the survey. Neither randomization of items nor adaptive questioning methods were used in this survey.

We excluded those who did not agree to participate in the study, those who were not registered with a team between January and March 2020, or those whose responses were incomplete or had missing values.”

We have also added Reference number 14:

Eysenbach, G. Improving the Quality of Web Surveys: The Checklist for Reporting Results of Internet E-Surveys (CHERRIES). J. Med. Internet Res. 2004, 6, e34. DOI:10.2196/jmir.6.3.e34.

Point 8: The results section is well structured and explained.

Response 8: Thank you very much for your comment.

Point 9: Bibliography should be added in the discussion. It is not sufficiently contrasted with other studies to give solidity to the conclusions.

Response 9: In accordance with your comment, we have revised the text of the Discussion

From

“the change in sleeping time was only 8 min, although it was significantly increased com-pared to that before the pandemic. The National Sleep Foundation in America has rec-ommended 9–11 hours of sleep daily for school-aged children [15]. Reduced physical ac-tivity and prolonged screen time have been associated with poor physical and mental health [5–7] and poor sleep [5,8]. Further prospective cohort studies are necessary to investigate the association between the physical and mental conditions of youth baseball players.”

To (Pages 6-7, Lines 196–208)

“Youth athletes in the United States reported significantly increased average sleeping time per night, from 7.9 hours before the pandemic to 9.4 hours during the pandemic [10]. Notably, the National Sleep Foundation in America has recommended 9–11 hours of sleep daily for school-aged children [18]. In a study with 160 student athletes, those with less than eight hours of sleep per night were reportedly 1.7 times more likely to sustain a musculoskeletal injury [19]. The reason for the difference in the changes in sleeping time between the previous study and this study may be that the participants of this study routinely slept for 9 hours per night even before the pandemic, which was longer than youth athletes in the United States. Moreover, there would be no reason to prolong the sleeping time because it was already optimal even before the pandemic. Further prospective cohort studies are necessary to investigate the lifestyle changes, especially prolonged RST, which affects the physical and mental conditions among youth baseball players.”

Additionally, we have added the following paragraphs:

(Page 7, Lines 225–241)

“Youth athletes in the United States reportedly trained for 9.7 hours per week on average before the pandemic; this number reduced significantly to 6.4 hours per week during the pandemic [10]. Although we could not directly compare the training time because such information was unavailable in our study, we posit that the lack of team practice may have detrimentally influenced the youth baseball players. Because baseball is a team sport, throwing and practicing in the bullpen requires at least one other player. In actuality, the incidence of injuries among Major League Baseball players after the COVID-19 pandemic was almost double of that before the pandemic; the increase in the injury incidence was seen in the upper extremities, spine/core, and other injuries, and not the lower extremities [20]. One of the reasons for the increased injury incidence was the insufficient throwing practice time [20] and throwing in fewer preseason innings [21]. On the other hand, the excessive number of throws [22] and training hours per week [23] were also reported as one of the factors of baseball-related pain in the shoulder and elbow among youth baseball players. Therefore, further investigation is required to clarify whether these changes in baseball team practice time after the pandemic was a factor in the increased incidence of injuries, especially in the upper extremities, among youth baseball players.”

(Page 7, Lines 251–258):

“Among Major League Baseball players, there were no significant differences in the incidence of injury to the lower extremity between the period before and during the pandemic, which was thought to be the result of isolated exercises and weight training using body-weight muscle strength without any fitness equipment/facilities [20]. In our study, we could not clarify the effect of voluntary exercise; however, we expect voluntary exercise is largely beneficial in terms of promoting physical activities, skill training, and prevention of injuries, especially in the lower extremities, among youth baseball players.”

We have also added Reference numbers 19, 20, 21, and 22:

Milewski, M.D.; Skaggs, D.L.; Bishop, G.A.; Pace, J.L.; Ibrahim, D.A.; Wren, T.A.L.; Barzdukas, A. Chronic Lack of Sleep Is Associated with Increased Sports Injuries in Adolescent Athletes. J. Pediatr. Orthop. 2014, 34, 129–133. DOI:10.1097/BPO.0000000000000151.

Platt, B.N.; Uhl, T.L.; Sciascia, A.D.; Zacharias, A.J.; Lemaster, N.G.; Stone, A.V. Injury Rates in Major League Baseball During the 2020 COVID-19 Season. Orthop. J. Sports Med. 2021, 9, 2325967121999646. DOI:10.1177/2325967121999646.

Paul, R.W.; Omari, A.; Fliegel, B.; Bishop, M.E.; Erickson, B.J.; Alberta, F.G. Effect of COVID-19 on Ulnar Collateral Ligament Reconstruction in Major League Baseball Pitchers. Orthop. J. Sports Med. 2021, 9, 23259671211041359. DOI:10.1177/23259671211041359.

Lyman, S.; Fleisig, G.S.; Waterbor, J.W.; Funkhouser, E.M.; Pulley, L.; Andrews, J.R.; Osinski, E.D.; Roseman, J.M. Longitudinal Study of Elbow and Shoulder Pain in Youth Baseball Pitchers. Med. Sci. Sports Exerc. 2001, 33, 1803–1810. DOI:10.1097/00005768-200111000-00002.

Point 10: The conclusions are presented clearly and are justified by the results presented. However, it is necessary to improve the discussion by including more references.

Response 10: Thank you very much for your comment. In accordance with your suggestion, we have revised the Discussion and included relevant references. Please see Response 9 for further details.

Point 11: The authors must incorporate the doi in the line reference 263: doi: 10.18053/jctres.06.202002.003

Response 11: In accordance with your comment, we have added the DOI to the reference:

Gagliardi, A.G.; Walker, G.A.; Dahab, K.S.; Seehusen, C.N.; Provance, A.J.; Albright, J.C.; Howell, D.R. Sports Participation Volume and Psychosocial Outcomes Among Healthy High School Athletes. J. Clin. Transl. Res. 2020, 6, 54–60. DOI:10.18053/jctres.06.202002.003.  (Page 9, Line 334-336)

Reviewer 4 Report

An interesting study, however, a small research group cannot be treated in terms of impact, but only cautious dependencies.

In the introduction, in my opinion, there is no theoretical basis, an explanation of what a lifestyle is and why such and not other components of this style have been taken into account by researchers.

The authors of the discussion emphasized that most teams suspended training, which should also be considered in this study as a confounding variable and why this happened. In the discussion, it might be worth referring to the results of studies conducted in other countries in the same or similar sports.

Author Response

Response to Reviewer 4 Comments

Point 1: An interesting study, however, a small research group cannot be treated in terms of impact, but only cautious dependencies.

Response 1: Thank you very much for your comment. We agree with your comment and also considered the sample of this study to be small. Hence, we have described it as a limitation of this study in the manuscript. Furthermore, in accordance with your comment, we have added the following sentence to the Conclusion (Page 9, Line 279-281):

”The results and conclusions of this investigation may be considered a basis and starting point for further research.”

Point 2: In the introduction, in my opinion, there is no theoretical basis, an explanation of what a lifestyle is and why such and not other components of this style have been taken into account by researchers.

Response 2: There is no established definition for the term “lifestyle” . In accordance with your comment, we added the sentence to explain why we chose "screen time," "study time," and "sleep time" as the lifestyle in this study:

(Page 2, Lines 47-50)

“The school closures made youth study at their home instead of going to school and they lost the regularity of getting up and going to the school at a certain time. It was also possible that the loss of time spent commuting to school is causing them to sleep longer. “

(Page 2, Lines 60-61)

“Therefore, youth lifestyle behavior should be focused on screen time, study time, and sleeping time.”

Moreover, we did not analyze other factors in the lifestyle because we did not collect data regarding these in this study. Therefore, we have added the following sentence to the Limitations (Page 8, Lines 267-270):

”Finally, data regarding the lifestyle other than those discussed in this study and participants’ physical activities that were not related to baseball were not collected, and differences in other lifestyle factors and non-baseball activity during the state of emergency compared to before the pandemic were not analyzed in this study.”

Point 3: The authors of the discussion emphasized that most teams suspended training, which should also be considered in this study as a confounding variable and why this happened. In the discussion, it might be worth referring to the results of studies conducted in other countries in the same or similar sports.

Response 3: We did not refer to the previous studies because we thought it would be more appropriate to discuss the reason for the cancellations based on the situation in Japan at that time. In accordance with your comment, we have added the following paragraphs to the Discussion (Page 7, Lines 217-225):

“We did not know the reason why these baseball teams canceled team practice, because we did not collect data regarding the underlying reason. However, the answers to questionnaire of this study were collected during the first state of emergency. This was the period when the Japanese people were confused by the high level of infection and its high mortality rate, and the Japanese government ordered school closures and restricted access to sports centers, playgrounds, and public parks. Therefore, it was not surprising that team practice was canceled. Even for the teams that had been practicing, the practices were arranged as short programs for a duration of 3 hours with infection control.”

Round 2

Reviewer 3 Report

Dear Authors,

I have reviewed the new version of the manuscript and the suggested changes have been made. 

Thank you very much.